# Host-Feeding Preference and Diel Activity of Mosquito Vectors of the Japanese Encephalitis Virus in Rural Cambodia

**DOI:** 10.3390/pathogens10030376

**Published:** 2021-03-21

**Authors:** Sébastien Boyer, Benoit Durand, Sony Yean, Cécile Brengues, Pierre-Olivier Maquart, Didier Fontenille, Véronique Chevalier

**Affiliations:** 1Medical and Veterinary Entomology Unit, Institut Pasteur du Cambodge, 5 Boulevard Monivong, Phnom Penh 12201, Cambodia; sonyyean168@gmail.com (S.Y.); pomaquart@pasteur-kh.org (P.-O.M.); didier.fontenille@ird.fr (D.F.); 2Laboratory for Animal Health, Epidemiology Unit, French Agency for Food, Environmental and Occupational Health and Safety (ANSES), University Paris-Est, 94701 Maisons-Alfort, France; benoit.durand@anses.fr; 3MIVEGEC Unit, Institut de Recherche pour le Développement (IRD), Université de Montpellier, CNRS, BP 64501, 34394 Montpellier, France; cecile.brengues@ird.fr; 4Epidemiology and Public Health Unit, Institut Pasteur du Cambodge, 5 Boulevard Monivong, Phnom Penh 12201, Cambodia; veronique.chevalier@cirad.fr; 5International Center of Research in Agriculture for Development (CIRAD), UMR AS TRE, 34090 Montpellier, France

**Keywords:** Japanese encephalitis virus, mosquitoes, host-feeding preference, *Culex vishnui*, Cambodia

## Abstract

Japanese Encephalitis (JE) is the most important cause of human encephalitis in Southeast Asia, and this zoonosis is mainly transmitted from pigs to human by mosquitoes. A better understanding of the host-feeding preference of Japanese encephalitis virus (JEV) major vectors is crucial for identifying risk areas, defining bridge vector species and targeting adapted vector control strategies. To assess host-feeding preference of JE vectors in a rural Cambodian area where JE is known to circulate, in 2017, we implemented four sessions of mosquito trapping (March, June, September, December), during five consecutive nights, collecting four times a night (6 p.m. to 6 a.m.), and using five baited traps simultaneously, i.e., cow, chicken, pig, human, and a blank one for control. In addition, blood meals of 157 engorged females trapped at the same location were opportunistically analyzed with polymerase chain reaction (PCR), using cow, pig, human, and dog blood primers. More than 95% of the 36,709 trapped mosquitoes were potential JE vectors. These vectors were trapped in large numbers throughout the year, including during the dry season, and from 6 p.m. to 6 a.m. Despite the apparent host-feeding preference of *Culex vishnui*, *Cx. gelidus*, and *Cx. tritaenhyorhincus* for cows, statistical analysis suggested that the primary target of these three mosquito species were pigs. Dog blood was detected in eight mosquitoes of the 157 tested, showing that mosquitoes also bite dogs, and suggesting that dogs may be used as proxy of the risk for human to get infected by JE virus.

## 1. Introduction

Japanese encephalitis (JE) is a vector borne zoonosis and one of the world’s leading encephalitic diseases, particularly in the Asia-Pacific region [1]. The region hosts more than three billion people and the annual incidence of JE is estimated at about 67,900 cases in 24 countries [2]. JE is caused by the Japanese encephalitis virus (JEV) belonging to the Flaviviridae family [3]. The historically described JEV cycle involves water birds as wild reservoir, pigs as amplifying hosts [4,5,6], and mosquito species as vectors [7,8]. Domestic birds are suspected to be secondary reservoir hosts [9,10,11]. Proximity to rice fields and pig rearing, particularly backyard farming, have been identified as major risk factors of JE in humans [12,13]. JE is thus considered a rural disease. However, several studies in Cambodia [14], Hong-Kong [15], Japan [16], Malaysia [17], Taiwan [18], Thailand [19], and Vietnam [20] have shown that JEV and JEV vectors can be found in peri-urban areas. Humans, horses and cattle are considered dead-end hosts [7,21,22]. 

There are at least 25 mosquito species playing a role in the transmission of JEV [23]. In Asian countries, *Cx. tritaeniorhynchus*, *Cx. vishnui*, *Cx. fuscocephala*, *Cx. gelidus*, *Cx. whitmorei* and *Mansonia uniformis* are considered the main vector species of JEV [21,24,25,26,27,28,29,30,31]. These mosquito species are widely distributed and highly abundant in SoutheastAsia [32,33,34]. They are also known for their opportunistic and generalistic trophic behavior [32,35,36,37,38].

*Culex tritaeniorhynchus* is considered the main vector of JEV transmission in Asia [28,29,30,39]. This mosquito species breeds mainly in the flooded rice fields [19,35,40,41,42]. *Culex gelidus* is described as the most important vector of JEV in India, Malaysia and Indonesia [33,43,44]. This species is able to breed in fresh and polluted water, and likely feeds on cattle [32,42]. The *Cx. pipiens* complex mosquitoes such as *Cx. pipiens pipiens, Cx. quinquefasciatus, Cx. pipiens molestus* considered as anthropophilic and urban mosquitoes are also described as JEV vectors [1,45]. 

In Cambodia, JEV was isolated for the first time in 1965 from *Cx. tritaeniorhynchus* mosquito species in Phnom Penh capital [40]. Most Cambodians live in areas considered endemic due to the wide distribution of JEV vectors and pig farming areas. However, data on JEV vectors in Cambodia are scarce. A first study based on the seroconversion of sentinel pigs in Kandal province, on the outskirts of the capital Phnom Penh, showed an intensive JEV circulation during both dry and rainy seasons [14]. In this study, over 99% of trapped mosquitoes were identified as potential JEV vectors, namely *Cx. tritaeniorhynchus*, *Cx. gelidus*, *Cx. vishnui,* and *Cx. quinquefasciatus* [14]. A second study investigated the population dynamics of different mosquito vector species mentioned above and showed a high relative abundance throughout the year with a significant abundance peak at the end of the rainy season [46]. Lastly, in Cambodia again, JEV genotype I was detected in *Cx. tritaeniorhynchus* [47].

“The host-feeding pattern of a mosquito population is the distribution of feeds taken on different vertebrate hosts” [48]. Genetically determined factors, environment and host availability impact feeding behavior of *Culex* mosquitoes [49,50]. Although pigs are central in the JE transmission, there is evidence that JEV virus can circulate in low pig density areas, suggesting the existence of secondary reservoirs such as domestic birds [4,10,11]. In these areas, the maintenance of JE could be facilitated by a multi-host system composed of pigs, ducks, chickens and vector populations exhibiting opportunistic feeding patterns [11].

A better understanding of these multi-hosts systems is needed to identify the main areas at risk. Our study aims at evaluating the respective feeding preferences of the main JE vectors in a rural Cambodian area where JE is endemic, and identify vector species that can be considered as bridge vectors between animal reservoirs and humans. To achieve this goal we implemented two complementary studies. In 2017, we first settled four sessions of mosquito trapping, during five consecutive nights, and using five baited traps simultaneously, i.e., cow, chicken, pig, human and a blank one for control. In addition, in 2018, a molecular host-analysis was performed on 157 engorged mosquitoes collected from the same site in 2016.

## 2. Results

### 2.1. Results of Baited Traps

#### 2.1.1. Diversity of Mosquito Species and Relative Abundance

With baited-traps, at least 34 species of mosquitoes were captured, belonging to 8 genera: *Aedes*, *Anopheles*, *Armigeres*, *Culex*, *Ficalbia*, *Lutzia*, *Mansonia*, *Mimomyia* (Table 1). Only 550 male mosquitoes were trapped (1.5%) and were discarded from data analysis. *Anopheles* genus represented at least 15 species and 11.2% of the trapped mosquitoes (4,122/36,709). At least nine *Culex* species were identified representing 86.8% (31,866/36,709) of all mosquitoes. 

We also identified at least 15 species of *Anopheles*, representing 30.6% of the total number of mosquitoes in March and at least nine species (8.7%) in September. At least four and six species of *Culex* were identified in March and September respectively, representing 34.5% and 55.7% of the mosquitoes. *Culex* mosquitoes were predominant in June and December representing 95.4% and 97.2% of trapped mosquitoes respectively. Among these *Culex*, *Cx. vishnui* represented 76.6% and 86.5% respectively, *Cx. gelidus* 9.9% and 7.2%, and *Cx. tritaeniorhynchus* 8.3% and 3.1%. Within the *Culex* genus, the group of *Cx. vishnui* was predominant with 3745 mosquitoes (33.3%) in March, 8318 in June (76.6%), 3213 in September (70.2%) and 8666 in December (86.5%). As a whole, the 4 main trapped *Anopheles* species, i.e., *An. peditaeniatus, An. sinensis, An. Campestris*, and *An. barbirostris* respectively represented only 6.8%, 1.7%, 0.3%, and 0.2% of the total number of trapped mosquitoes.

Among the 34 identified species, six species are well-known JEV vectors: *Cx. bitaeniorhynchus*, *Cx. fuscocephala*, *Cx. gelidus*, *Cx. quinquefasciatus*, *Cx. tritaeniorhynchus,* and *Cx. vishnui* [24,25,26,27,28,29,30,31]. These six species represented 86.6% (31,786/36,709) of all the mosquitoes captured during the four entomological field missions, with high abundance levels (March: 68.8%, June: 95.4%, September: 87.8%, and December: 96.9%) (Table 1).

#### 2.1.2. Host-Feeding Preference and Biting Activity Pattern

The effects of bait type (chicken, cow, human, and pig), month, time of collection, and position of the trap were studied for the most abundant and presumably important species in terms of JEV transmission, namely *Cx. gelidus*, *Cx. tritaeniorhynchus* and *Cx. vishnui* (Table 2). 

The mosquito activity was highest from dawn, until 9 p.m. for *Cx. tritaeniorhynchus*, and midnight for *Cx. vishnui* and *Cx. gelidus* (Table 2; Figure 1). The biting activity of *Cx. tritaeniorhynchus* was maximum between 6 and 9 p.m. (OR = 0.6, 0.5 and 0.3, for 9 p.m.–0, 0–3 a.m. and 3–6 a.m. respectively). Compared to this trapping time slot, *Cx. vishnui* was significantly less active from 0 to 3 a.m. (OR = 0.7, *p* = 0.02) and from 3 to 6 a.m. (OR = 0.4, *p* < 0.0001) and *Cx. gelidus*, from 3 to 6 a.m. (OR = 0.2, *p* < 0.0001) (Table 2; Figure 1).

The number of trapped mosquitoes also varied significantly with the month of trapping (Table 2; Figure 2). *Cx. vishnui* was significantly more abundant in December and June than in March (OR = 0.6, *p* = 0.002) and September (OR = 0.3, *p* < 0.0001). *Cx. tritaeniorhynchus* was the most abundant in March (OR = 14.3, *p* < 0.0001) and less abundant in September (OR = 0.5, *p* = 0.008). *Cx. gelidus* was significantly less abundant in March (OR = 0.2, *p* < 0.0001) compared to the other months. 

Regarding the feeding preference, the three mosquito species were attracted by the four different host species (Figure 2). Whatever the mosquito species, individual-level models showed that, cow-and pig-baited traps were significantly more attractive than human-,chicken-and blank-baited traps (Table 2). Compared to pig, the odd-ratios were higher for cow (1.6 for *Cx. vishnui*, 1.4 for *Cx. tritaeniorhynchus* and 1.5 for *Cx. gelidus*) and always significantly less than 1 for human-, chicken-, and blank-baited traps (Table 2). 

However, these results changed when considering Body Surface Area (BSA) level models: pig- and chicken-baited traps were significantly more attractive than human-baited traps (Table 2). For a given BSA unit, *Cx. vishnui* and *Cx. gelidus* were more attracted by pigs and chickens, and *Cx. tritaeniorhynchus* by pigs. The difference between cow and pig was significant only for *Cx. gelidus* (OR: 0.6, *p* = 0.04). Although not significant, the same trend was observed for *Cx. vishnui* (OR = 0.7; *p* = 0.08) and *Cx. tritaeniorhynchus* (OR = 0.6; *p* = 0.07) (Table 2). 

Compared to Site 1, we caught more *Cx. gelidus* individuals on Site 2 (OR = 1.7; *p* = 0.03), 4 (OR = 3.1; *p* < 0.0001) and 5 (OR = 2.0; *p* = 0.003) (Table 2).

### 2.2. Blood Meal Analysis of Engorged Mosquitoes

PCR was performed on blood meals of female mosquitoes captured with light traps in the same area the year before the host-feeding preference experiment. In 2016, 157 engorged female mosquitoes were collected, representing at least seven mosquito species belonging to three genera (Table 3). A total of 118 individual mosquitoes were identified to species level and 41 mosquitoes were identified only to the genus level. Some specimens were not identified to the species level due to the quality of the field samples caught with light traps. The mosquitoes were *Anopheles sinsulaeflorumor/bangalensis* (*n* = 4), *Cx. gelidus* (*n* = 26), *Cx. quinquefasciatus* (*n* = 6), *Cx. tritaeniorhynchus* (*n* = 20), *Cx. vishnui* (*n* = 52), *Mansonia annulifera* (*n* = 4) and *Ma. uniformis* (*n* = 6) (Appendix A). Surprisingly, none of the PCR realized with human primers and chicken primers were positive. Sixty eight individuals were engorged with cow blood (43.3%), 47 with pig blood (29.9%), and 8 mosquitoes (*Cx. gelidus*, *Cx. tritaeniorhynchus*, *Cx. vishnui* and *Cx. sp*) were engorged with dog blood (5.1%) (Table 3). More specifically, *Cx. gelidus* (17/24) and *Cx. quinquefasciatus* (4/6) were mainly found with cow blood, while *Cx. tritaeniorhynchus* and *Cx. vishnui* were found mainly with pig or cow blood (Table 3). Finally, blood meals of 34 mosquitoes (21.7%) were taken from hosts that could not be identified with our primers.

## 3. Discussion

Previous entomological studies performed in rural and peri-urban Cambodian areas showed a high mosquito species diversity [46,51]. Our results confirm the presence of a significant number of mosquito species present around human habitats. In terms of number of species, the genus *Anopheles* has the highest species diversity in our study. This genus is known to be very diverse in Southeast Asia [52]. The presence of cultivation, in particular rice paddies, as well as the presence of many rivers, also known to be breeding sites frequented by *Anopheles* can explain this high diversity. *Culex* genus is ultra-dominant and represents almost 90% of mosquitoes caught in baited traps, with *Cx. vishnui*, *Cx. gelidus* and *Cx. tritaeniorhynchus*. *Cx. vishnui* has already been described as predominant in Cambodia [46] and is known to be involved in the transmission of JEV even if no vector competence study has yet demonstrated it formally. *Cx vishnui* was detected positive with JEV in India, Malaysia and India [33,53,54,55,56]. JEV was isolated from *Cx. gelidus* in India [33,57] and in pig farms in Malaysia [58]. JEV was isolated from *Cx. tritaeniorhynchus* in Java [59], Indonesia [60], India [57], Vietnam [23], Malaysia [58], Taiwan [61], and Cambodia [47]. Interestingly, the fourth most abundant species is *Anopheles peditaeniatus* in which, also, JEV was already isolated in India [57]. These four species represented 93% of the 36,709 trapped mosquitoes. 

*Cx. vishnui*, *Cx. gelidus* and *Cx. tritaeniorhynchus* species were trapped throughout the year, suggesting that JEV can circulate even during the cold season, from November to February. The baited double net trap methodology was used with success to trap mosquito’s vector of *Plasmodium* sp. in Lao PDR [62] and Cambodia [63]. In addition, the vectors were active during each stretch of the night, including the first one from 6 to 9 p.m. In Cambodia, rural inhabitants are active (cooking, dining, showering) outside, generally under the stilt house until 9–10 p.m. This critical timeframe could be a preferential moment for exposure to JEV vectors. Indeed, *Culex* mosquitoes are classically described as exophilic and nocturnal mosquitoes [48]. Consequently, sleeping under a mosquito net is useful but not enough to protect humans against JE transmission.

The feeding behavior of mosquitoes is dependent on host preference and host selection [64,65] and influenced by several factors such as carbon dioxide, host-skin volatiles and compound blends in the specific case of host seeking [66]. Host selection is defined as the feeding pattern in nature, represented by the relative frequency of different blood meal sources of a mosquito population in time and space. In the present survey, host selection is assessed using the individual-based model. Host preference is defined as the trait to preferentially select a particular vertebrate host as a food source, over the other species that are equally available [64]: this was assessed using the BSA-level model. Although the host preference is determined by numerous intrinsic physiological characteristics of the vector [67,68,69], host selection is primarily influenced by ecological [65] and chemical factors such as fatty acids, n-aliphatic carboxylic acids, lactic acid playing an important role in differential olfactory attraction [66]. Both results are important in terms of host-feeding preference.

Our trapping results confirm that *Culex* are opportunistic mosquitoes. The 3 species of concern were attracted by the four baits. However, and as expected, we observed a significant difference in conclusion between the two statistical analyses. Whatever the mosquito species, individual-level models showed that, cow- and pig-baited traps were significantly more attractive than human-, chicken- and blank-baited traps. Indeed, in the field, we observe a greater number of mosquitoes around a cow than a pig, mainly for its significant release of heat and carbon dioxide. Even if CO_2_ was certainly the best attractant for mosquito host location, especially for zoophilic mosquitoes [70,71], the skin volatile compounds of the different species, especially for pig and cow, certainly plays a more specific attractant role at short distances as already demonstrated on another *Culex* species, Culex *quinquefasciatus* [72].

According to BSA-level model, *Cx. vishnui* and *Cx. gelidus* species are primarily attracted by pigs and chicken. This result is consistent with the literature describing *Cx. vishnui* having pigs and birds as preferred hosts [73,74], but being able to feed on cow and man in the absence of its main hosts [75]. In Taiwan, *Cx. vishnui* was reported to feed on pigs [76] and in Thailand on buffalo and cattle [26]. For *Cx. gelidus*, this same generalist, opportunistic behavior with a zoophilic preference was observed during our study. This result is confirmed in the literature, citing it as a highly general mosquito with a preference for cow and pig [77]. It should also be noted that this mosquito is considered to be zoophilic although it can conveniently feed on humans [78]. For *Cx. tritaeniorhynchus*, the BSA-level model shows a strong preference for pigs then cows and chickens, and then humans. These zoophilic preferences of mosquitoes for cows and pigs have been described since 1959 [73]. In the absence of cattle, the species is attracted to human but is slow to feed, whereas in the presence of cattle, man is almost completely ignored [73]. 

Results of blood meal PCR analyses confirm the statistical analyses performed on trapping data, despite the very low number of mosquitoes. PCR results also showed that the main potential JEV vector species in Cambodia can feed on dogs. Experimentally, infected dogs showed very low viremia suggesting that dogs are not involved in JEV circulation [79]. However, dogs live alongside humans and, as suggested by the results of a serological survey carried out in the same area [11], they could be a relevant indicator of the risk for humans to get infected by JEV. In Kandal province where the present survey was carried out, the dog/human ratio was estimated to be 1:4 on average (V. Chevalier, pers.com on 20 January 2021). Since dogs are supposed to be a dead end host, they may participate with cows to a zooprophylactic effect in areas where the ratio dog/human is high, and to some extent reduce mosquito predation on humans.

Despite the existence of human vaccines, JE remains a major public health problem in Southeast Asia, especially in children. In Cambodia, the opportunistic behavior of JE vectors may facilitate JE circulation within a multi-host system, especially in low-pig densities areas. Estimating the respective role of these vectors according to their environment will be an important step to better understand this multi-host system, to refine the identification of areas at risk and improve prevention, and control strategies in the future.

## 4. Materials and Methods

### 4.1. Study Area

The study was conducted in Kbal Chhroy village, Porti Ban commune, Koh Thom District, Kandal province, Cambodia (11.219846° N, 105.039502° E, WGS 84 system). The study site was located in a house backyard, near the Bassac River, and a pig farm (around 350 located at 200 m, on the other side of the road). The surrounding landscape, mainly rural, was dominated by crops (mango, corn, beans etc.) and rice cultivation. The owner also reared pigs, cows, and chickens at the backyard of the household, located at around 20 m of the experiment.

### 4.2. Bait Trapping Survey

#### 4.2.1. Trapping

Adult mosquitoes were trapped in 2–6 March, 21–25 June, 11–15 September and 4–8 December 2017, using five double net baited traps: one cow, one pig, one human, eight chickens, and 1 empty trap for control. We used eight chickens to minimize a potential bias due to an increased CO_2_ attractiveness of pig/cow/human compared to chickens.

The trap consisted in a classic baited double net. One untreated mosquito net (size X = 450, Y = 286 cm, H = 220 cm) was raised slightly above the ground (around 30 cm) to allow mosquitoes to enter the trap from its base. Another untreated mosquito net (size X = 280, Y = 200 cm, H = 190 cm) was settled under the first one, protecting the bait from mosquito bites. The size of cage-trap was equal for all the treatments. The same animal was used during the entire night and for each session. The cow was partially encaged and tied as normally, the pig and the chicken were in their habitual cage. Animals were under the same standard conditions as usual.

The traps were used during five consecutive nights, with a turn-over of the baits at the different positions (Site 1 to Site 5, located 4 m from each other) designed as a *carre latin*. The area of each trap was cleaned every morning: removing of cow, pig and chicken faecal material and ground washed from with fresh water.

The baits were settled at 06.00 p.m., and mosquito collection was performed every 3 h at 09.00 p.m., 00.00 a.m., 03.00 a.m. and 06.00 a.m. by two technicians who caught mosquitoes trapped between the two nets using manual aspirators. Collected mosquitoes were killed with an insect spray and stored in an icebox at +4 °C.

#### 4.2.2. Mosquito Identification

Mosquitoes were identified on site under stereomicroscope (SZ61 Olympus, Tokyo, Japan) and using morphological mosquito identification keys of Southeast Asia countries [80,81,82,83,84,85]. After identification, mosquitoes were sorted in 1.5 mL Eppendorf tube (Thermo Scientific, Waltham, MA, USA) according to the trap site, date and hour of catch, bait type, mosquito species, blood fed status (engorged or not), and sex, with a maximum of 30 individuals per tube.

#### 4.2.3. Statistical Analysis

We focused the analysis on the well-known JEV vectors, namely *Cx. gelidus*, *Cx. tritaeniorhynchus* and *Cx. vishnui* [24,25,26,27,28,29,30,31], which were collected in more than 10% of capture sessions (a capture session being, here, a combination of a date, a collection time, a trap position and a bait type). For each species, we used a generalized linear model to analyze the variations of the number of captured mosquitoes according to the bait type (chicken, pig, cow, human or blank), the month (December, March, June or September), the collection time (6 p.m.–9 p.m., 9 p.m.–0 a.m., 0 a.m.–3 a.m., or 3 a.m.–6 a.m.), and the position of the trap (five distinct position names, i.e., Site 1 to Site 5).

Two error distributions are commonly used to model count data such as numbers of trapped mosquitoes: the Poisson distribution and the negative binomial distribution. Because of the strong overdispersion observed with Poisson distributions, we chose using negative binomial error distributions. When modelling count data, the use of an offset allows associating each observation with a level of “exposure”. The statistical model is then used to quantify the effect of the explanatory variables per unit of “exposure”. In our case, the “exposure” is the quantity of baits used. We fitted two groups of models, corresponding to two distinct measures of this quantity. In “individual-based” models, the offset was the logarithm of the number of individuals used as baits (8 for chicken and 1 for the 4 other baits). The estimated effect of the type of bait (i.e., species) then indicated the increase of the number of trapped mosquitoes induced by an additional individual of that species in the trap. In “body surface area (BSA)-based” models, we used the logarithm of the total BSA as an offset, based on the following values of BSA: 0.13 m^2^ for a chicken, 1.51 m^2^ for a pig, 3.45 m^2^ for a cow, and 1.81 m^2^ for a human [86,87] (blank trap collections were discarded for this part of the analysis). The estimated effect of bait type in this case corresponded to the increase of the number of trapped mosquitoes induced by an additional square meter of BSA of the species in the trap. Note that changing the measure of bait quantity (i.e., the offset) only affected the estimated effect of the bait type, not the month, time of collection, and trap position. Pigs were chosen as the reference (“Ref.”) for the model while ‘December’, ‘6 p.m.–9 p.m.’ and ‘site 1′ were arbitrary selected as reference by being respectively the first month of collection, the first quarter of the night session, and the first site.

All the statistical analyses were performed using R v. 3.6.3 [88], packages MASS version 7.3.53 [89], ggplot2 v. 3.3.2 [90], and cowplot v. 1.1.0 [91].

### 4.3. Analysis of Blood Fed Mosquitoes

#### 4.3.1. Blood-Fed Specimens

Blood fed mosquitoes were caught in the same area, Kbal Chhroy village, at the same household in 2015–2016 using light traps [46] and conserved at −20 °C. In 2018, all blood-fed females (*n* = 157) were analyzed with polymerase chain reaction (PCR), using specific primers for the most abundant animals living in the close vicinity of the household, namely pigs, human, cows, chickens, and dogs.

#### 4.3.2. DNA Extraction

Mosquitoes were separated in 1.5 mL Eppendorf tube with two iron beads, 200 µL of Cetyl TrimethylAmmonium Bromide (CTAB, Sigma-Aldrich, Saint-Louis, MO, USA) to grind the body for 4 min in the Tissuelyser II QIAGEN (Program 2, Fq 29). After a centrifugation of 15 sec at 12,000 rpm and a 65 °C water bath during 5 min, 200 µL of chloroform (Sigma-Aldrich) were added, and centrifuged again for 5 min. The supernatant was transferred in another 1.5 mL tube and we added 200 µL of isopropanol, mixed and centrifuged for 15 min. Then, we removed the isopropanol, add 200 µL of ethanol 70%, centrifuged for 5 min, and removed ethanol. After drying the tubes, we added 20 µL of H_2_O pure water for PCR. All samples were stored in the freezer at −20 °C after a final DNA CTAB extraction [92].

#### 4.3.3. Polymerase Chain Reaction Assay

Two different PCRs were used to identify the animal origin of blood meals: one multiplex able to detect human, cow, pig and dog blood, and a uniplex for chicken [93,94]. The 25 µL reaction mixture contained 1x buffer, 2.5 mM MgCl_2_, 0.2 mM dNTP, 10 pmoles UNREV 1025, 5 pmoles of Dog 368F primers, Human 741F primers, Cow 121F primers, Pig 573F primers, and 0.5 U of Taq DNA polymerase with H2O, and DNA diluted 1/10 [93]. The amplification conditions adapted from Kent et al. [93] were a 1st step at 94 °C for 5 min, a 2nd step of 35 cycles of 30 s at 94 °C, 1 min at 58 °C and 1 min at 72 °C, and a final step at 72 °C for 10 min. Chicken 470F primers were used alone [94]. Polymerase Chain Reaction (PCR) products were analyzed using a 2.5% agarose gel (Tris Acetate EDTA) and visualized under U.V. (Molecular Imager, Bio-Rad, Hercules, CA, USA) [93]. Ethidium bromide-stained agarose gel showed whole blood of mosquitoes fed on each blood source. Control products amplified whole blood of chicken (290 bp), human (350 bp), pig (400–500 bp), cow (600–700 bp) and dog (700–800 bp).

## Figures and Tables

**Figure 1 pathogens-10-00376-f001:**
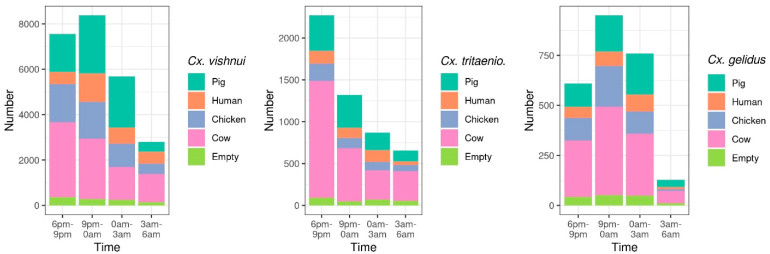
Mosquito abundance of the three main species *Culex vishnui*, *Culex tritaeniorhynchus* and *Culex gelidus* on different baits (pig, human, chicken, cow and empty) at different time of collection during the night (every 3 h from 6 p.m. to 6 a.m.). Y-axis have different scales.

**Figure 2 pathogens-10-00376-f002:**
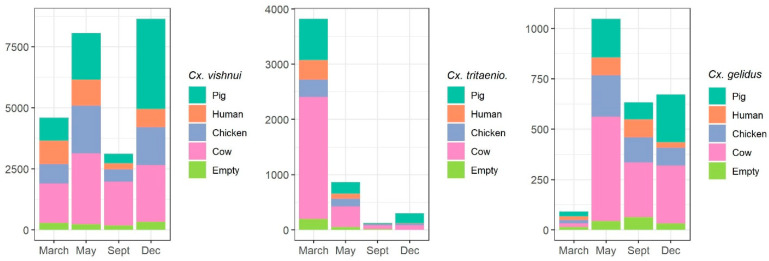
Mosquito abundance of the three main species *Culex vishnui*, *Culex tritaeniorhynchus* and *Culex gelidus* on different baits (pig, human, chicken, cow and empty) during the 4 trapping months (March, June, September and December). Y-axis represents the total number of collected mosquitoes. Y-axis have different scales.

**Table 1 pathogens-10-00376-t001:** List and total number of mosquitoes species caught by double nets with animal (cow, chicken, pig) and human baited traps during 5 consecutive nights for each month. The study was conducted from March to December 2017 in Kbal Chroy village in Cambodia. In bold, confirmed mosquito species vector of Japanese encephalitis virus.

Mosquito Species	March	June	September	December	Total
*Aedes aegypti*	2		1		3
***Aedes albopictus***			**1**	**4**	**5**
*Aedes* sp.	1	3		1	5
Total Aedes	3	3	2	5	13
*Anopheles barbirostris*	1		87		88
*Anopheles barbumbrosus*	1	23	13	5	42
*Anopheles campestris*	32	2	77		111
*Anopheles crawfordi*	2		2		4
*Anopheles donaldi*	1		1		2
*Anopheles hodgkini*	2			1	3
*Anopheles indefinitus*	1	4	26		31
*Anopheles insulaeflorum*	2				2
*Anopheles nigerrimus*	2	2			4
*Anopheles nitidus*	2			6	8
*Anopheles peditaeniatus*	2320	18	93	80	2511
*Anopheles pursati*	2			2	4
*Anopheles roperi*	1				1
*Anopheles sinensis*	518	30	51	14	613
*Anopheles sintonoides*	8				8
*Anopheles* sp.	543	38	47	62	690
Total Anopheles	3438	117	397	170	4122
*Armigeres kesseli*			1		1
***Armigeres subalbatus***	**6**	**1**	**17**	**53**	**77**
*Armigeres* sp.	4	3			7
Total Armigeres	10	4	18	53	85
***Culex bitaeniorhynchus***		**3**	**2**		**5**
***Culex fuscocephala***			**5**	**1**	**6**
***Culex gelidus***	**88**	**1074**	**667**	**719**	**2548**
*Culex nigropunctatus*		1	6	2	9
***Culex quinquefasciatus***	**39**	**25**	**2**	**15**	**81**
***Culex sitiens***		**2**			**2**
***Culex tritaeniorhynchus***	**3869**	**897**	**130**	**308**	**5204**
***Culex vishnui***	**3745**	**8318**	**3213**	**8666**	**23,942**
*Culex* sp.	2	40		27	69
Total Culex	7743	10,360	4025	9738	31,866
*Ficalbia* sp.	1				1
Total Ficalbia	1	0	0	0	1
*Lutzia fuscana*				2	2
Total Lutzia	0	0	0	0	2
*Mansonia annulifera*	1		13	10	24
*Mansonia bonneae*	2				2
*Mansonia indiana*			6		6
*Mansonia uniformis*	52	354	99	34	539
*Mansonia* sp.		22		2	24
Total Mansonia	55	376	118	46	595
*Mimomyia luzonensis*			10	3	13
*Mimomyia* sp.		3		2	5
Total Mimomyia	0	3	10	5	18
unknown genus	0	2	5	0	7
Total	11,250	10,865	4575	10,019	36,709
Species number	25	15	23	18	34

**Table 2 pathogens-10-00376-t002:** Result of the negative binomial generalized linear models with the number of trapped mosquitoes (*Cx. vishnui*, *Cx. tritaeniorhynchus* and *Cx. gelidus*) per trapping session as the outcome and bait type, collection time and position of trap as explanatory variables.

Mosquito Species	Variable	Value	Individual-Level Model		BSA ^b^-Level Model	
			Odds-Ratio	*p*-Value	Odds-Ratio	*p*-Value
			(95% CI ^a^)		(95% CI)	
*Culex vishnui*	Bait	Pig	Ref.		Ref.	
		Human	0.5 (0.4–0.8)	0.002	0.4 (0.3–0.7)	0.001
		Chicken	0.09 (0.06–0.1)	<0.0001	1.1 (0.7–1.6)	0.67
		Cow	1.6 (1.1–2.4)	0.016	0.7 (0.5–1.1)	0.08
		Empty	0.2 (0.1–0.3)	<0.0001	NA	
	Month	December	Ref.			
		March	0.6 (0.4–0.8)	0.002		
		June	1.0 (0.7–1.5)	0.9		
		September	0.3 (0.2–0.4)	<0.0001		
	Hour	6 p.m.–9 p.m.	Ref.			
		9 p.m.–0 a.m.	1.0 (0.7–1.4)	0.9		
		0 a.m.–3 a.m.	0.7 (0.5–0.9)	0.02		
		3 a.m.–6 a.m.	0.4 (0.2–0.5)	<0.0001		
*Culex tritaeniorhynchus*	Bait	Pig	Ref.		Ref.	
		Human	0.2 (0.1–0.4)	<0.0001	0.2 (0.1–0.3)	<0.0001
		Chicken	0.05 (0.03–0.08)	<0.0001	0.5 (0.3–1.0)	0.03
		Cow	1.4 (0.8–2.4)	0.2	0.6 (0.4–1.1)	0.07
		Empty	0.2 (0.1–0.3)	<0.0001	NA	
	Month	December	Ref.			
		March	14.3 (8.6–23.7)	<0.0001		
		June	4.6 (2.7–7.9)	<0.0001		
		September	0.5 (0.3–0.9)	0.008		
	Hour	6 p.m.–9 p.m.	Ref.			
		9 p.m.–0 a.m.	0.6 (0.4–1.0)	0.03		
		0 a.m.–3 a.m.	0.5 (0.3–0.9)	0.01		
		3 a.m.–6 a.m.	0.3 (0.2–0.5)	<0.0001		
*Culex gelidus*	Bait	Pig	Ref.		Ref.	
		Human	0.5 (0.3–0.8)	0.005	0.4 (0.3–0.7)	0.0003
		Chicken	0.09 (0.06–0.15)	<0.0001	1.1 (0.7–1.7)	0.7
		Cow	1.5 (0.9–2.3)	0.08	0.6 (0.4–1.0)	0.04
		Empty	0.3 (0.2–0.5)	<0.0001	NA	
	Month	December	Ref.			
		March	0.2 (0.1–0.3)	<0.0001		
		June	1.6 (1.1–2.4)	0.01		
		September	1.1 (0.7–1.7)	0.7		
	Hour	6 p.m.–9 p.m.	Ref.			
		9 p.m.–0 a.m.	1.3 (0.9–1.9)	0.2		
		0 a.m.–3 a.m.	1.1 (0.8–1.7)	0.5		
		3 a.m.–6 a.m.	0.2 (0.1–0.3)	<0.0001		
	Position ^c^	Site 1	Ref.			
		Site 2	1.7 (1.0–2.7)	0.03		
		Site 3	1.3 (0.8–2.0)	0.3		
		Site 4	3.1 (1.9–4.9)	<0.0001		
		Site 5	2.0 (1.3–3.2)	0.003		

^a^: Confidence interval, ^b^: Body Surface Area, ^c^: Position of the trap.

**Table 3 pathogens-10-00376-t003:** Blood-feeding analysis of blood-fed mosquitoes analyzed by PCR.

Mosquito Species (N = 157)	Dog	Pig	Cow	Negative
*Anopheles sinsulaeflorumor/bangalensis*	0	1	3	0
*Anopheles* sp.	0	0	0	1
*Culex gelidus*	1	5	17	1
*Culex quinquefasciatus*	0	0	4	2
*Culex* sp.	3	17	11	9
*Culex tritaeniorhynchus*	1	8	9	2
*Culex vishnui*	3	15	18	16
*Mansonia annulifera*	0	1	6	3
Total	8	47	68	34

## Data Availability

The data presented in this study are available on request from the corresponding author.

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
