# Peer review of "Host-Feeding Preference and Diel Activity of Mosquito Vectors of the Japanese Encephalitis Virus in Rural Cambodia"

_pathogens, 2021, doi:10.3390/pathogens10030376_

Round 1

Reviewer 1 Report

This manuscript aimed to assess host-feeding preference of JE vectors and identify bridge vectors between animal reservoirs and humans for JEV in a rural Cambodian area. This study carried out the mosquito surveillance in a household rearing pigs, cow, and chicken. They analyzed the mosquito species attracted by different animal baits, and found the potential JEV vectors were dominant. The primary target of JEV vectors was cows or pigs while different models were used for the statistical analysis. However, in terms of host-feeding preference, the authors only analyzed 157 of blood fed mosquitoes. Overall, this manuscript presented weak evidence for host-“feeding” preference of JEV vectors.

Major comments

  1. Authors should independently analyze female and male mosquitoes or focus on female mosquitoes, which are responsible for transmission of JEV to human.
  2. I think it was inappropriate to use “Host-feeding preference” in this manuscript since there was no evidence showing bait-captured mosquitoes fed on the bait.
  3. What’s the implication for the different results between“individual-based” models and body surface area (BSA)-based” models?
  4. Sample size of blood-feeding mosquitoes was small.

Minor comments

  1. Line 27: I couldn’t find these results for Cx. quinquefasciatus
  2. Line 29-30: There was no evidence to suggest dogs could develop viremia after infection and be the amplifying host for JEV.
  3. Line 88: remove “in 2018”
  4. Table 1: The columns of “Potential” and “Confirmed” were not necessary. Move the row of species number right below the row of mosquito species. Why were some mosquito species highlighted in bold?
  5. Line 119: What does entomological missions mean here?
  6. Line 121: please include the calculation for biting activity in Materials and Methods.
  7. Figure 1. Please indicate what y-axis is and re-order Dec. after Sept.
  8. Line 267-268: Why did you select March, May, September, and December?

Reviewer 2 Report

This is generally a well-written manuscript with interesting and useful results. I suggest a few changes and clarifications. With some small modifications, I find it worthy of publication.

The abstract and introduction are sound. The information provided is accurate and appropriately researched and referenced. 

There a few grammatical discrepancies, for example:

Page 2, Line 76  - "there are evidences" should be: "there is evidence"

Page 4, Line 139 - "hosts' species" should be: "host species"

Page 6, Line 166 - "some specimen" should be: "some specimens"

Page 7, Line 223 - "carbon dioxyde" should be: "carbon dioxide"

Page 7, Line 246 - "to be dead end host" should be: "to be a dead end host"

Page 8, Line 269 - "eight chicken" should be: "eight chickens"

Page 8, Line 270- "compared to chicken" should be: "compared to chickens"

The results are clearly presented. I think that a reader would benefit from a graphical depiction of the diel activity of these potential vectors, as I find this information to be some of the most relevant in terms of choosing and implementing control measures. Right now, that info is buried in the tables. I would suggest making these findings more prominent in the text and abstract as well. Seasonal results (also important) are already presented this way. Similar bar graphs for diel activity would complement this well and provide much important information at a glance. These observations are also noted in the discussion, but again, they are important enough to receive further highlighting.

I would have also liked to have seen data on the diel abundance of engorged specimens as this sheds light on the timing of actual feeding behavior. Although, the pattern might not be different from that seen in actual collections, it would have been worth checking.

In the discussion, the rational for greater attraction of mosquitoes to the cow over other hosts is given as greater output of CO2 and heat. It should be noted that CO2 has generally been shown to be an activator of host-seeking activity, while heat and other factors tend to be more important in terms of relative attractiveness. This varies by species. I would suggest that the authors consider the role of volatile substances (fatty acids, n-aliphatic carboxylic acids, lactic acid) in differential olfactory attraction. This should be added as part of the discussion to fully account for the factors at play.

Overall, the methodology of this paper was fairly sound. I would point out that some of the choices were rather arbitrary. For example, having eight chickens per baited trap versus one of the other larger animal or human baits, did not have a firm basis of equivalence, particularly since the authors did not seem to be fully aware that factors other than respiratory output and heat might be responsible for differences in attractiveness. It's good that they included a blank as a control.

Reviewer 3 Report

The manuscript "Host-feeding preference and diel activity of mosquito vectors of the Japanese encephalitis virus in rural Cambodia" reports the findings of mosquito trapping study at a single location using a number of potential live baits (human, pig, cow, chicken, empty). The findings of the study are clearly presented and conclusions supported by the data. There are a number of key changes needed and a number of text corrections required (listed below).

Major comments:

  1. The Introduction states that the study could "provide recommendations for surveillance and control in a country where JEV is endemic." Based on the findings of this study, what were they?
  2. For Table 2, state in the legend and Materials and methods how "Ref." was determined.

Minor corrections

Line 29. Change to "8 mosquitoes of 157 tested, .." (if correct)

Line 93. State in section 2.1.1. that all data refers to female mosquitoes.

Line 126. change to "highest from dawn , until 9pm" (if correct)

Line 145.  note in the legend that the y-axis have different scales.

Line 148. first reference to BSA-level, define this acronym.

Line 162. state at this point that light traps were used to capture blood-fed mosquitoes.

Line 190. rephrase sentence "Culex vishnui was detected JEV positive ...."

Line 201. change to "... including between 6 and 9 pm. In Cambodia, rural inhabitants are active (cooking, dining, showering)  outside,.."  "..moment for exposure to JEV vectors."

line 223. "carbon dioxide" , what are 'ground-realities'?

Line 239. "despite being performed".

Line 242. "..dogs live alongside humans .." 

Line 244. How would infection be determined?

Line 247. suggest ".. to some extent reduce mosquito predation on humans.."

Line 251/2. repetition of low-pig densities.

Line 260. Not sure household is the correct term in  this context.

Line 270. CO2, 2 lower case.

Line 282. suggest "...chicken faecal material washed from the ground with fresh water."

Line 337.  Ethanol E lower case.

Line 342.  'uniplex' rather than simplex.

Line 343. MgCl2 2 lower case.

Round 2

Reviewer 1 Report

Please response all the comments.

Author Response

Dear Reviewer,

We are deeply sorry and sincerely apologized. We did all the changes and commented them. Sincerely,

the authors
